# Determinants of obstructed labour and its adverse outcomes among women who gave birth in Hawassa University referral Hospital: A case-control study

Melaku Desta[1]*, Zenebe Mekonen[2], Addisu Alehegn Alemu[1], Minychil Demelash[3], Temesgen Getaneh[1], Yibelu Bazezew[1], Getachew Mullu Kassa[1], Negash Wakgari[4]

1 Department of Midwifery, College of Health Science, Debre Markos University, Debre Markos, Ethiopia,
2 Department of Obstetrics and Gynecology, School of Medicine, Hawassa University, Hawassa, Ethiopia,
3 Department of Midwifery, College of Health Science, Wachamo University, Wachamo, Ethiopia,
4 Department of Midwifery, College of Medicine and Health Science, Ambo University, Ambo, Ethiopia

* Melakd2018@gmail.com

## Abstract

### Background

Globally, obstructed labour accounted for 22% of maternal morbidities and up to 70% of perinatal deaths. It is one of the most common preventable causes of maternal and perinatal mortality in low-income countries. However, there are limited studies on the determinants of obstructed labor in Ethiopia. Therefore, this study was conducted to assess determinants and outcomes of obstructed labor among women who gave birth in Hawassa University Hospital, Ethiopia.

### Methods

A hospital-based case-control study design was conducted in Hawassa University Hospital among 468 women. All women who were diagnosed with obstructed labour and two consecutive controls giving birth on the same day were enrolled in this study. A pretested data extraction tool was used for data collection from the patient charts. Multivariable logistic regression was employed to identify determinants of obstructed labor.

### Results

A total of 156 cases and 312 controls were included with an overall response rate of 96.3%. Women who were primipara [AOR 0.19; 95% CI 0.07, 0.52] and multigravida [AOR 0.17; 95% CI 0.07, 0.41] had lower odds of obstructed labour. While contracted pelvis [AOR 3.98; 95% CI 1.68, 9.42], no partograph utilization [AOR 5.19; 95% CI 1.98, 13.6], duration of labour above 24 hours [AOR 7.61; 95% CI 2.98, 19.8] and estimated distance of 10 to 50 kilometers from the hospital [AOR 3.89; 95% CI 1.14, 13.3] had higher odds. Higher percentage of maternal (65.2%) and perinatal (60%) complications occurred among cases (p-value < 0.05). Obstructed labour accounted for 8.3% of maternal deaths and 39.7% of stillbirth.

**Data Availability Statement:** All relevant data are within the manuscript and its Supporting Information files.

**Funding:** The author(s) received no specific funding for this work.

**Competing interests:** The authors have declared that no competing interests exist.

**Abbreviations:** HUCSH, Hawassa University Comprehensive Specialized Hospital; LMICs, Low- and Middle-Income Countries; MM, Maternal Mortality; OL, Obstructed Labour; SNNPR, Southern Nations Nationalities and People's Region.

Uterine rupture, post-partum haemorrhage and sepsis were the common adverse outcomes among cases.

## Conclusion

Parity, contracted pelvis, non-partograph utilization, longer duration of labour and longer distance from health facilities were determinants of obstructed labour. Maternal and perinatal morbidity and mortality due to obstructed labour are higher. Therefore, improvement of partograph utilization to identify complications early, birth preparedness, complication readiness and provision of timely interventions are recommended to prevent such complications.

## Background

Globally, more than 303,000 women die every year from pregnancy and childbirth-related causes. Millions of women also suffer from complications related to pregnancy and childbirth like haemorrhage, hypertensive disorders and obstructed labor. For example, in 2015, direct obstetric causes of maternal mortality (MM) accounted for about 86% of all maternal deaths globally [1, 2]. One of the direct obstetric causes of MM is obstructed labour (OL), which is the failure of descent of the fetus in the birth canal for mechanical reasons despite good uterine contractions [3].

Obstructed labour is one of the leading causes of maternal and perinatal morbidity and mortality. Despite a rapid drop in global maternal death in the last decades, obstructed labour is still considered a significant challenge [2, 4]. It has a negative economic impact in developing countries due to long hospitalization and scarce resources budgeted for the healthcare system [1, 5–7]. Obstructed labour affects 3 to 6% of labouring women in developing countries [8].

Obstructed labour is responsible for 22% of obstetrical complications, 9% of all maternal deaths in low- and middle-income countries (LMICs). In sub-Saharan Africa (SSA) countries, OL is responsible for 24% of maternal deaths. It is also associated with 9% of maternal and perinatal mortality [9]. The burden of OL in Ethiopia is estimated to be 11.79% [10]. However, the prevalence varies across different regions, 46% in Debre Markos hospital [11], 17.5% in Tigray region [12] and 9.6% in Adama hospital, Oromia region [13]. In Ethiopia, OL is associated with 17% of maternal death and 38.08% of still birth based on recent study [14] and 36% of maternal death when combined with uterine rupture [15]. As the result, the issue of OL and maternal and perinatal survival was one of the main focuses of Sustainable Development Goals (SDGs) [16].

Many of the morbidities and deaths due to obstructed labour are preventable and treatable. However, studies showed that the burden of obstructed labour and its adverse maternal and perinatal outcomes appear to be high and remain a common challenge in Ethiopia [13, 17–23]. Different studies conducted across the countries showed that there were different determinants of obstructed labor such as, maternal age, maternal residence, women's education status, women's occupational status [24], distance from the hospital /health center, parity, antenatal visit, weeks of gestation at the first visit of antenatal care [24], age at first birth, fetal presentation, history of pregnancy-related complications and birth weight [13, 19, 25].

Therefore, identification of determinants and outcomes of obstructed labour is essential for the reduction of morbidities and mortality associated with OL. There are limited studies conducted in Ethiopia on the determinants of OL and its adverse outcomes. Therefore, this study was conducted to assess the determinants and adverse outcomes of obstructed labour in Hawassa University comprehensive specialized hospital, Southern Ethiopia.

## Methods

### Study design, setting and population

Unmatched case-control study was conducted in Hawassa University Comprehensive Specialized Hospital (HUCSH), Hawassa city, the capital of Southern Nations Nationalities and People's Region (SNNPR). Hawassa city is located 275 km south of Addis Ababa (capital city of Ethiopia). HUCSH is one of the largest hospitals in the region and serves as a specialized and teaching hospital. The hospital is offering a full range of comprehensive emergency obstetric care services. The average numbers of births were around 12,456 in the 3 years period from January 1, 2015, to August 31, 2017.

All women who gave birth in HUCSH in the last 3 years before the data collection period were considered as a source population. Whereas, randomly selected women who gave birth in the last 3 years and fulfilled the inclusion criteria were the study population. Cases were women who were diagnosed to have OL by the most senior person (resident and obstetrician), and controls were women who had no obstructed labour in the hospital on the same day as enrolled cases regardless of their mode of delivery. All women who gave birth after 28 weeks of gestation or weight of at least 1000 gm were included in the study. Cases and controls were selected after reviewing of women's chart, delivery logbook and operation notes. However, women who gave birth with a scheduled cesarean section were not included in this study.

### Sample size and sampling procedure

Openepi version 3.01 software was used to calculate the 3 sample size using double population proportion formula, on the assumption of case to control ratio of 1:2, 95% confidence level, Power of 80% and least extreme odds ratio of 2.00 and the sample size is calculated based on for the first objective/ determinants by considering rural resident as determinant factor of OL according to a study done in Ethiopia making the calculated sample size was 329, but by considering 10% nonresponse rate, 363 sample size was estimated. For the adverse maternal and perinatal outcomes, a study done in Uganda [26] as maternal complication and perinatal mortality as adverse birth outcome of OL was used, making the largest sample size of 486 sample (S1 File). Thus, his study included a total of 486 women (162 women for cases and 324 women for controls). For cases, the delivery chart of women who gave birth in the hospital in the last three years was randomly selected. For controls, two women were selected after each case. All women's charts were retrieved from the hospital record office and were cross-checked with the delivery logbook and operating theatre registers.

### Variables and measurements

The dependent variable of this study was obstructed labour. Whereas, the independent variables were categorized as socio-demographic factors, obstetric, health facility and fetal factors. The sociodemographic factors included in this study were age, residency and specific district), and obstetrical factors were parity, previous cesarean section, previous stillbirth, antenatal care utilization, gestational age, membrane status and pelvic status. Health facility factors included were partograph follow up, distance from the health facility, duration of labour and source of referral. Fetal factors include were malpresentation, malposition, and weight of the newborn.

Obstructed labour is the failure of descent of the fetus in the birth canal for mechanical reasons despite good uterine contractions [3]. In addition, the inadequate pelvis was diagnosed when the medical team leader (the residents or obstetricians and gynaecologists) assessed the labouring woman and confirms as feto-pelvic disproportion secondary to the contracted pelvis. A contracted pelvis is defined as a pelvis in which one or more of the pelvic diameters are

reduced below the normal and that can interfere with the normal mechanism of labour. It is diagnosed using internal pelvimetry such as the sacral promontory is felt easily, or interspinous diameter is touched by 2 examining fingers simultaneously, or bituberous diameter cannot admit the closed fist of the hand or the ischial spines is prominent or the coccyx is not mobile.

### Data collection procedure and quality control

Data were collected by using a pretested data extraction tool by reviewing the obstetric records of women who gave birth. Admission history, labour follow up sheet, delivery summary, antenatal care (ANC) follow up sheet and operation notes were used. The data extraction tool was adapted from different related kinds of literatures [18, 21], and was modified to assess the determinants and adverse outcomes of obstructed labour. The questionnaire was prepared in the English language. Two days of training were given for the data collectors and supervisors on the objectives of the study and ways of data collection. Five BSc midwives as data collectors and one MSc Clinical midwife supervisor were recruited in this study. Collected data were checked on daily for completeness and consistency. Three days of training were given for the data collectors and supervisors, focusing on the objective of the study and data collection process.

### Data processing and analysis

Data were checked, cleared and entered on Epi Data version 3.1 software and exported to Statistical Package for Social Science (SPSS) software version 20 for further analysis. The proportion of the cases and controls were computed. Variables in bivariable logistic regression with p-value < 0.25 were entered into multivariable logistic regression. Model fitness was checked using Hosmer and Lemeshow goodness of fit test statistics, and it showed that the model was fitted, p-value = 0.46. After the regression analysis, variables with a p-value < 0.05 were used as statistically significant factors and odds ratio (OR) with 95% confidence interval (CI) were used to measure the strength of association. Maternal and perinatal outcomes of obstructed labour were also examined.

## Results

### Sociodemographic and prenatal characteristics

A total of 156 out of 162 cases (96.3% and 312 out of 324 controls (96.3%) were included. The Mean age of the women was 26.9 years (SD ± 5.6). In addition, 64% of cases and 217 (69.6%) of controls were in the age group of 20–34 years. Almost 77% of cases and 56.1% of controls reside outside Hawassa. Similarly, 59% of cases were Oromo ethnic group, and fifty-nine (37.8%) of cases were grand-multiparous. Likewise, 66% of cases and 236 (76.4%) of controls had antenatal care visits during the current pregnancy (**Table 1**).

### Intrapartum, fetal and health facility-related characteristics

Twenty eight percent of cases had contracted pelvis and 124 (91.1%) of cases had ruptured membranes during labour after admission to the hospital. The progress of labour among 62 (39.7%) of cases and 87 (27.9%) of controls were not monitored using partograph. Nearly 58% of cases were admitted to the hospital for more than 24 hours during labour. Seventy-four (47%) of cases were referred to the hospital from other health institutions and 121 (38.8%) of controls were self-referred (**Table 2**). Cephalopelvic disproportion (38.5% vs. 11.5%) and malpresentation (32.3% vs. 19.9%) were common among cases than controls, respectively.

**Table 1. Sociodemographic and antenatal characteristics of women who gave birth in HUCSH from 2015–2017.** Sociodemographic and prenatal characteristics of participants in HUCSH, 2018.

| Variables | Category | Case (%) | Control (%) | P-value |
|---|---|---|---|---|
| Age | < 20 year | 30 (19.2) | 48 (15.4) | 0.056 |
| | 20–34 year | 100 (64.1) | 217 (69.6) | |
| | ≥ 35 year | 26 (16.4) | 47 (15.1) | |
| Residence | Outside Hawassa | 120 (76.9) | 175 (56.1) | 0.004 |
| | Hawassa | 36 (23.1) | 137 (43.9) | |
| Ethnicity | Sidama | 45 (28.8) | 144 (46.2) | 0.34 |
| | Oromo | 92 (59) | 124 (39.7) | |
| | Amhara | 17 (10.9) | 30 (9.6) | |
| | Others | 2 (1.3) | 14 (4.5) | |
| Parity | One | 41 (26.3) | 114 (36.5) | 0.0001 |
| | 2–4 | 56 (35.9) | 144 (46.2) | |
| | ≥ 5 | 59 (37.8) | 54 (17.3) | |
| Previous scar | Yes | 40 (34.8) | 71 (35.9) | 0.54 |
| | No | 75 (65.2) | 127 (64.1) | |
| Previous stillbirth | Yes | 21 (18.3) | 10 (5.1) | 0.27 |
| | No | 94 (81.7) | 188 (94.9) | |
| Diabetes in recent pregnancy | Yes | 8 (5.1) | 16 (5.1) | 0.086 |
| | No | 148 (94.9) | 296 (94.9) | |
| ANC visit | Yes | 103 (66) | 236 (76.4) | 0.035 |
| | No | 53 (34) | 73 (23.6) | |
| Frequency of ANC | < 4 visit | 62 (60.2) | 103 (42.9) | 0.001 |
| | ≥ 4 visit | 41 (39.8) | 133 (57.1) | |

Others—Gurage, Gedeo, wolayita

Similarly, 59% of cases and 33.6% of controls were delivered through cesarean section, and 35.9% of cases were delivered by laparatomy (Fig 1). During laparotomy, total abdominal hysterectomy (TAH) was done for 39 (25.7%) of the cases, subtotal hysterectomy was performed for 3 (2%) of cases, and 6 (3.9%) of the cases had uterine repair with bilateral tubal ligation (BTL).

## Determinants of obstructed labour

In bivariate analysis, 11 variables were significant and were fitted for multivariable logistic regression with a p-value of < 0.25. After controlling of confounding effect, only 6 variables (parity, pelvic status, partograph utilization, delay of seeking care and estimated distance from the facility) were the significant determinants of OL (Table 3). Primiparous women were 81% times [AOR = 0.19, 95% CI: 0.07, 0.52] and multigravida women were 83% times [AOR = 0.17, 95% CI: 0.07, 0.41] less likely to have OL than grand multiparous women. Similarly, women who had contracted pelvis were about 4 times more likely to have the chance of obstructed labour than those who had adequate pelvis [AOR = 3.98, 95% CI: 1.68, 9.42]. Moreover, women whose progress of labour was not monitored with partograph were five times more likely to encounter OL than their counterparts [AOR = 4.93, 95% CI: 0.76, 13.7]. The odds of OL was 7.61 times [AOR = 7.61, 95% CI: 2.98, 19.8] higher among women who had a longer duration of labour (> 24 hrs) before reaching the hospital than those reaching the hospital < 12 hours. The odds of OL were 3.89 times more likely among women who reside within 10–50 kilometers estimated distance from the facility than those who reside below 10 kilometers distance [AOR = 3.89, 95% CI: 1.14, 13.3].

**Table 2. Intrapartum and health facility factors of obstructed labour among women who gave birth in HUCSH from 2015–2017.** Intrapartum, and health facility characteristics of obstructed labour in HUCSH, 2018.

| Variables | Category | Cases (%) | Controls (%) | P-value |
|---|---|---|---|---|
| Gestational age | < 42 week | 126 (83.7) | 239 (85.7) | 0.054 |
| | ≥ 42 week | 21 (16.3) | 40 (14.3) | |
| Pelvis status | Contracted | 44 (28.2) | 35 (11.2) | 0.004 |
| | Unknown | 28 (17.9) | 36 (11.5) | |
| | Adequate | 84 (53.8) | 241 (77.2) | |
| Fetal membrane status | PROM | 14 (9) | 39 (12.5) | 0.001 |
| | Rupture in labour | 142 (91.1) | 121 (38.8) | |
| | Intact | 0 | 152 (48.7) | |
| Sex of newborn | Male | 108 (69.2) | 182 (58.3) | 0.46 |
| | Female | 48 (30.8) | 130 (41.7) | |
| Birth weight | < 4000 g | 119 (76.3) | 278 (87.4) | |
| | ≥ 4000 g | 37 (23.7) | 40 (12.8) | 0.02 |
| Partograph utilization | Yes | 13 (8.3) | 140 (44.9) | 0.0001 |
| | Unknown | 81(51.9) | 85 (27.2) | |
| | No | 62 (39.7) | 87 (27.9) | |
| Duration of labour | > 24 hr | 90 (57.7) | 62 (19.9) | 0.02 |
| | 12–24 hr | 50 (32.1) | 103 (33) | |
| | < 12 hr | 16 (10.2) | 147 (47.1) | |
| Source of referral | Self | 21 (13.5) | 121 (38.8) | 0.37 |
| | Healthcenter | 61 (39.1) | 117 (37.5) | |
| | Hospital | 74 (47.4) | 74 (23.7) | |
| Estimated distance from home to facility | < 10 km | 17 (10.9) | 130 (41.7) | 0.007 |
| | 10–50 km | 46 (29.5) | 79 (25.3) | |
| | > 50 km | 93 (59.6) | 103 (33) | |

## Maternal and perinatal adverse outcomes of obstructed labour

Almost 65% of women who had OL developed at least one form of maternal complications when compared with 56 (17.9%) among women who had no obstructed labour, which accounts for 8.3% of the case fatality ratio (*p-value < 0.05*). The most common morbidities among women who had OL were long hospital admission (48.9%), uterine rupture (38.5%), post-natal anemia (37.8%), PPH (29.5%) and sepsis (14%), *p*-value < 0.05. A perinatal complication occurred among 60% of cases and 40% of the controls. Of those, 39.7% of cases and 19.6% of controls had stillbirths, and 20.6% of cases and 23% of the controls had a low Apgar score (**Table 4**).

## Discussion

The study assessed the determinants of obstructed labour and its adverse outcomes in southern Ethiopia. Accordingly, different factors that affect the occurrence of obstructed labour were identified. Lower birth order was a protective factor of obstructed labour. In contrary to this finding, previous studies done in Nigeria [27], Rwanda [28], Uganda [26] and Sudan [29] revealed that primiparity was associated with OL. A study conducted in LMICs [30] also showed that gravidity ≥ 2 was protective of OL. This variation might be due to socio-demographic differences and more risk of malpresentation and malposition among primigravida women. Moreover, those women who have lower gravidity may utilize maternal health services than grand multipara women. Hence, women with lower birth order utilize skilled birth

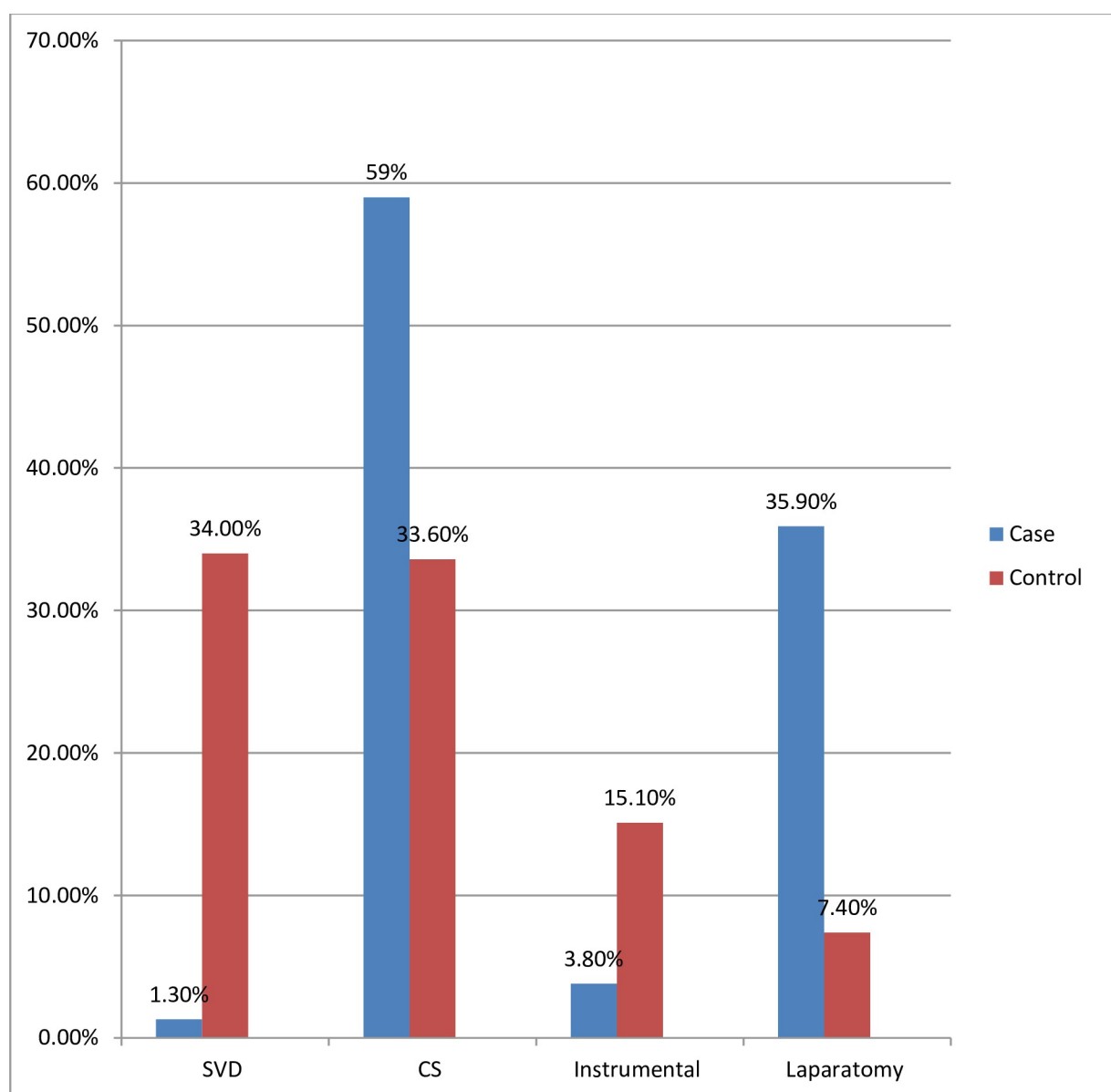

**Fig 1. Management options of obstructed labour in HUCSH, Ethiopia 2018.**

attendants earlier than women who had higher birth order and this could consequently improve the health care seeking ability of the woman to prevent obstructed labour. Additionally, it might be also due to higher odds of obesity and macrosomia among women with higher birth order due to decreased levels of physical activity and higher energy intake [31–34]. Moreover, obesity could directly increase the risk of fetal macrosomia [32, 35–37].

Women who had contracted pelvis were more likely to develop OL than women with the adequate pelvis. This finding is supported by other studies [38, 39]. This might be due to mechanical obstruction of the passage of the fetus due to an ill fit between maternal pelvic dimensions and neonatal size at delivery and poor fetal head-to-cervix contact. This might be due to the high burden of malnutrition in childhood in Oromia and SNNPR [40, 41]. Stunting causes a small, flattened pelvis, and being obese in the later life and development of her

**Table 3. Determinants of obstructed labour in Hawassa referral Hospital, among women who gave birth in HUCSH from 2015–2017.** Determinants of obstructed labour in HUCSH, southern Ethiopia, 2018.

| Variables | Category | Cases | Controls | COR [95%CI] | AOR [95%CI] |
|---|---|---|---|---|---|
| Resident | Outside Hawassa | 120 | 175 | 2.6 (1.69, 4.03)* | 0.43 (0.17, 1.03) |
| | Hawassa | 36 | 137 | 1 | 1 |
| Gravidity | One | 41 | 114 | 0.33 (.19, 0.55)* | 0.19 (0.07, 0.52)֍ |
| | 2–4 | 56 | 144 | 0.35 (0.22, 0.58)* | 0.17 (0.07, 0.41)֍ |
| | ≥ 5 | 59 | 54 | 1 | 1 |
| ANC visit | yes | 103 | 236 | 0.6 (0.39, 0.92)* | 1.03 (0.04, 25.3) |
| | No | 53 | 73 | 1 | 1 |
| Frequency of ANC | < 4 visit | 103 | 62 | 1.99 (1.25, 3.18)* | 1.23 (0.61, 2.46) |
| | ≥ 4 visit | 137 | 41 | 1 | 1 |
| Gestational age | < 42 week | 126 | 239 | 1.0041(0.06, 1.34) | 0.39 (0.02, 0.59) |
| | ≥ 42 week | 21 | 40 | 1 | 1 |
| pelvic status | Contracted | 44 | 35 | 3.61 (2.17, 5.99)* | 3.98 (1.68, 9.42)֍ |
| | Unknown | 28 | 36 | 2.23 (1.28, 3.88)* | 1.79 (0.67, 4.79) |
| | Adequate | 84 | 241 | 1 | 1 |
| Sex | Male | 108 | 182 | 1.6 (1.07, 2.42)* | 0.92 (0.45, 1.88) |
| | Female | 48 | 130 | 1 | 1 |
| Birth weight | 2500–4000 g | 119 | 278 | 0.47 (0.39, 1.05)* | 0.71 (0.31, 1.63) |
| | ≥ 4000 g | 37 | 40 | 1 | 1 |
| Partograph utilization | Yes | 13 | 140 | 1 | 1 |
| | Unknown | 81 | 85 | 10.2 (5.38, 19.5)* | 4.93 (0.76, 13.7) |
| | No | 62 | 87 | 7.67 (3.98, 14.7)* | 5.19 (1.98, 13.6)֍ |
| Duration of labour | > 24 hr | 90 | 62 | 13.4 (7.25, 24.5)* | 7.61 (2.98, 19.8)֍ |
| | 12–24 hr | 50 | 103 | 4.46 (2.41, 8.26)* | 1.39 (0.57, 3.39) |
| | < 12 hr | 16 | 147 | 1 | 1 |
| Source of referral | Self | 21 | 121 | 0.17 (0.09, 0.31)* | 1.06 (0.32, 3.58) |
| | Healthcenter | 61 | 117 | 0.52 (0.33, 0.82)* | 1.33 (0.59, 2.98) |
| | Hospital | 74 | 74 | 1 | 1 |
| Estimated distance | < 10 km | 17 | 130 | 1 | 1 |
| | 10–50 km | 46 | 79 | 4.45 (2.39, 8.29)* | 3.89 (1.14, 13.3)֍ |
| | > 50 km | 93 | 103 | 6.9 (3.87, 12.3) * | 3.89 (0.91, 16.6) |

* Shows variables selected for multi variable logistic regression at p< 0.25

֍ Significant factors at p-value < 0.05

offspring, might make OL genetically predisposed [42]. Hence, improvements in maternal and child nutrition are essential to prevent OL and improve reproductive outcomes [43–45].

This study also showed that the absence of partograph utilization was significantly increased OL. This is supported by different studies in Ethiopia [19, 25]. This might because partograph helps the health care provider in identifying the slow progress of labour and provides an early warning system for early referral and may also help to initiate appropriate interventions. Hence, proper partograph utilization improves labour outcomes and reduces obstructed labour [46–49]. The study also demonstrated that the odds of OL were higher among women with longer duration of labour (more than 24 hours) before arrival to the health facility than women with shorter duration (less than 12 hours). This finding is consistent with the study done in Oromia, Ethiopia [13]. This might be due to the fact that delay of health-seeking care is known factor of OL due to absence of appropriate timely interventions of prolonged labour or abnormal labor.

**Table 4. Maternal and perinatal outcomes of obstructed labour in Hawassa University specialized Hospital, Southern Ethiopia.** Maternal and perinatal outcomes of obstructed labour in Hawassa University specialized Hospital, Southern Ethiopia.

| Variables | Cases (%) | Controls (%) | P-value |
|---|---|---|---|
| Maternal complication | 102 (65.2) | 56 (17.9) | 0.01 |
| Maternal death | 13 (8.3) | 6 (1.9) | 0.035 |
| Post partum haemorrhage | 64 (29.5) | 13 (4.2) | 0.045 |
| Sepsis | 22 (14) | 16 (5.3) | <0.0001 |
| uterine rupture | 60 (38.5) | 28 (9) | 0.023 |
| Bladder rupture | 3 (1.9) | 1(0.03) | 0.06 |
| Post-natal anemia | 59 (37.8) | 38 (12.2) | 0.17 |
| Shock | 33 (21.2) | 21(6.7) | 0.25 |
| Fistula | 13 (8.3) | 4 (1.3) | <0.001 |
| Transfusion | 37 (23.7) | 31 (9.9) | 0.18 |
| Hysterectomy | 52 (33.3) | 28 (9) | 0.0057 |
| Long hospital admission | 70 (48.9) | 63 (21.4) | 0.039 |
| Perinatal complication | 94 (60) | 128 (40) | 0.0035 |
| Stillbirth | 72 (46.2) | 51 (16.3) | 0.023 |
| Low Apgar score | 57 (23.1) | 19 (20.6) | 0.35 |
| NICU admission | 12 (12.5) | 27 (10) | 0.46 |

In addition, the study also indicated that the higher odds of obstructed labour among women who reside within 10–50 kilometers compared to those who reside below 10 kilometers. This finding is in line with studies done in Tanzania [50] and Ethiopia [24, 25]. This might be due to the fact that women living close to hospitals get life-saving obstetric information and services in labour earlier, reduce delays from referral and treatment, and reduce maternal morbidity.

The study also assessed the adverse maternal and perinatal outcomes among cases and controls. Accordingly, nearly two-thirds of women with OL encountered at least one form of adverse maternal outcomes. This finding is higher than studies done in Nigeria [27], Uganda and Mizan Tepi, Ethiopia [51]. The possible variation might be due to delays in referral and treatment of OL, prolonged labour, study setting, sample size and methodological differences between the studies. However, the finding of this study is lower than studies done in Bangladesh [52], India [53, 54], Suhul hospital, Ethiopia [20], and Metu Karl hospital, Ethiopia [18]. This might be due to the commitment of the hospital to improve maternal healthcare provision, safe surgery with the senior obstetrician and EMONC service.

Besides, this study showed that OL resulted in 8.3% of maternal deaths among cases. This figure is higher than a study done in India, [53, 54], Uganda, [26], Sudan, [29], Nigeria, [55], Tanzania, 2% [50], Bangladesh, [52] and Ethiopia, [20]. However, the findings of the current study were lower than a similar study in Sudan, [56]. This might be due to the high burden of morbidity (uterine rupture, severe anaemia, postpartum haemorrhage and sepsis) among cases, delay in referral and treatment and variation in the study setting. Because the current study was conducted in a tertiary hospital and the number of referred cases may be higher. Moreover, improved diagnosis, transfer, and treatment for OL reduce the rate of maternal mortality [57] by preventing the progression of prolonged labour to OL. Additionally, one-third of women with cases did not get ANC service, therefore, prevents getting birth preparedness and complication readiness (BPCR) intervention. Previous studies conducted in Ethiopia also showed a low percentage of BPCR in Oromia, [58] and SNNPR, [59].

Uterine rupture was also the commonest adverse maternal outcome among cases in the current study than controls. This could be because of prolonged duration of labour, higher

previous cesarean section and multiparty among cases than controls. Moreover, above half of women with cases had a longer duration of labour above 24 hours, 34% had previous CS and 56% of women were multiparous in the current study. As the duration of labour increases, the uterus becomes exhausted and the uterine muscle loses its integrity mainly for multiparous and previous CS. This leads to uterine rupture when the condition is exacerbated by a delay in receiving care due to a longer distance from clinical facilities. This is supported by other studies in 40 low-income countries [9], Ethiopia [18–20, 60], Uganda [26] and Sudan [29]. Postpartum anemia is higher in case of obstructed labour due to antepartum and postpartum haemorrhage when it is encountered with uterine rupture [60].

Furthermore, the findings of this study showed that stillbirth was the commonest adverse perinatal outcome among cases (39.7%). This is in line with studies done in Suhl Hospital, Ethiopia [20], Metu Karl hospital, Ethiopia [18] and Sudan [29]. This is likely attributed to difficulties in delivering the fetus during caesarean section. Because the fetal head is impacted in the pelvis and needs a longer operation time. The highest proportion of maternal morbidity, intrapartum asphyxia, delay of referral and lower ANC visit, limited BPCR results in adverse perinatal outcomes. Hence, stillbirth is related with maternal morbidity [61–63] and mortality [64, 65]. But, it is lower than studies done in Pakistan [66] and Ethiopia [19]. This might be due to variation in the study setting, study period and improvements of the care provision. Thus, the provision of a timely maternal and perinatal continuum of care should be an area of improvement to reduce stillbirth.

The study has certain strengths and limitations. Due to the use of a case-control study design, the study was able to determine causal relations between the outcome variable and independent variables and included a relatively larger sample size. However, the findings of this study should be interpreted with some inevitable limitations. The retrospective nature of the study might prevent data collection for some variables like educational level, type of delays, socioeconomic status, nutritional status, and infrastructure of the health facilities as these variables were not registered in women's obstetric cards. There might be also subjectivity in the diagnosis of OL and in estimating distance from home to health facility. Additionally, the study might underestimate adverse perinatal outcomes. Because, the study was unable to assess some perinatal outcomes mainly neonatal death due to neonatal intensive care unit (NICU) admission and after discharge.

## Conclusions

Parity, contracted pelvis, partograph utilization, duration of labour and longer distance from the health facility was significantly associated with obstructed labour. Obstructed labour increased maternal and perinatal morbidity and mortality. Prolonged admission, uterine rupture, post-partum haemorrhage and sepsis were the commonest adverse outcomes of obstructed labour. Encouraging the use of family planning, improving partograph utilization, birth preparedness and complication readiness plan, early referral, diagnosis of OL is recommended. Additionally, community mobilization on the need of complication readiness plan and training for healthcare providers on prevention of obstructed labour at all health facilities is essential.

## Supporting information

**S1 File. The sample size determination for determinants and adverse outcomes of obstructed labour.**
(DOCX)

**S2 File. The STROBE statement for determinants of obstructed labour.**
(DOCX)

**S1 Data.**
(SAV)

## Acknowledgments

Authors are thankful for HUCSH workers.

## Author Contributions

**Conceptualization:** Melaku Desta, Zenebe Mekonen, Minychil Demelash, Yibelu Bazezew, Negash Wakgari.

**Data curation:** Melaku Desta, Addisu Alehegn Alemu, Temesgen Getaneh, Yibelu Bazezew, Getachew Mullu Kassa, Negash Wakgari.

**Formal analysis:** Melaku Desta, Temesgen Getaneh, Negash Wakgari.

**Funding acquisition:** Minychil Demelash, Getachew Mullu Kassa, Negash Wakgari.

**Investigation:** Addisu Alehegn Alemu, Minychil Demelash, Temesgen Getaneh.

**Methodology:** Melaku Desta, Zenebe Mekonen, Addisu Alehegn Alemu, Minychil Demelash, Temesgen Getaneh, Yibelu Bazezew, Negash Wakgari.

**Project administration:** Zenebe Mekonen, Addisu Alehegn Alemu, Temesgen Getaneh, Negash Wakgari.

**Resources:** Melaku Desta, Zenebe Mekonen, Yibelu Bazezew, Negash Wakgari.

**Software:** Melaku Desta, Yibelu Bazezew.

**Supervision:** Zenebe Mekonen, Addisu Alehegn Alemu, Minychil Demelash, Yibelu Bazezew, Getachew Mullu Kassa, Negash Wakgari.

**Validation:** Melaku Desta, Getachew Mullu Kassa.

**Visualization:** Melaku Desta, Negash Wakgari.

**Writing – original draft:** Melaku Desta.

**Writing – review & editing:** Zenebe Mekonen, Addisu Alehegn Alemu, Minychil Demelash, Temesgen Getaneh, Yibelu Bazezew, Getachew Mullu Kassa, Negash Wakgari.

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
