## [Decision Letter · Decision Letter 0]

14 Oct 2021

PONE-D-21-03875

Determinants of obstructed labor and its adverse outcomes among women who gave birth in Hawassa University Referral Hospital: A case control study

PLOS ONE

Dear Dr. Desta,

Thank you for submitting your manuscript to PLOS ONE. After careful consideration, we feel that it has merit but does not fully meet PLOS ONE’s publication criteria as it currently stands. Therefore, we invite you to submit a revised version of the manuscript that addresses the points raised during the review process.

Please respond to the reviewers' thoughtful comments. In particular, please include a description of all the determinants of obstructed labour reported previously in other studies in the background section; elaborate on the sample size; and rewrite the Discussion section. Please correct grammatical and spelling errors.

We look forward to receiving your revised manuscript.

Kind regards,

Nancy Beam, PhD

Staff Editor

PLOS ONE

Journal Requirements:

2. Thank you for including your ethics statement:  "The study protocol was reviewed and approved by Hawassa University, College of Medicine and Health Science Institutional Review Board (IRB) committee with a Ref. No IRB/163/10. Official letter of cooperation was obtained from College of Medicine and Health Science to Hawassa specialized Hospital and permission was secured from medical director. The consent was ***consent was waived by an IRB ***. All study participants charts were reviewed and returned after data extraction. Information obtained in the study was stored confidentially. ".   

Please amend your current ethics statement to include the full name of the ethics committee that approved your specific study / waived the need of consent.. 

Reviewers' comments:

Reviewer's Responses to Questions

**Comments to the Author**

1. Is the manuscript technically sound, and do the data support the conclusions?

Reviewer #1: Yes

Reviewer #2: Yes

2. Has the statistical analysis been performed appropriately and rigorously? 

Reviewer #1: Yes

Reviewer #2: Yes

3. Have the authors made all data underlying the findings in their manuscript fully available?

Reviewer #1: No

Reviewer #2: Yes

4. Is the manuscript presented in an intelligible fashion and written in standard English?

Reviewer #1: Yes

Reviewer #2: No

5. Review Comments to the Author

Reviewer #1: Good study on determiants of OL and its outcomes in Hawassa University Referral Hospital.

Abastract:

In the methods area mention should be made of the fact that the study was based on patient records/charts.

Background:

Authors ought to refer to a similar study conducted by Mulugeta et al. 2020. Reasons as to why the study was done in Hawassa?

Results:

Well presented though the tables appear vebe very long.

Discussion:

Please authors include freference to a study by Mululeta Dile et al. (2020) titled: Determinants of obstructed labour among women attending intrapartum care in Amhara Region NW Ethiopia: a hospital based unmatched case-control study.

In the the limitations there is no mention about missing data which is common iin use of secondary data.

Conclusion:

Strong recommendation on the use of family planning though relevant but does not come directly from the data.

General comment: authors should try and correct the spelling and grammatical errors.

Reviewer #2: 1. Generally, the article is very important to share with the wider community as it addresses determinants of obstructed

labour which is one of the direct causes of maternal deaths. Knowing the determinants of obstructed labour will help to put measures that will improve maternal health and ultimately reduce maternal mortality.

2. The abstract is more than 300 words based on journal guidelines, try to reduce redundant words.

3. There were a lot of grammatical errors throughout the documents. Needs to revise the whole document.

4. Background: The authors need to unpack and describe all the determinants of obstructed labour reported previously in other studies in the background section,.

5. Ethic: In Ethics statement line 4, correct repeated word, “consent was”

6. Sample size: The sample size calculation is not well computed; This is a case control study and decided to take a ratio of 1:2, how did you arrive with a total number of 486 women (162 women for cases and 324 women for controls)?. This should be well elaborated

7. Methods: In the Method section, under “Study design, setting and population” line 14..This sentence is not clear, “All women who gave birth after 28 weeks of gestation or weight of at least 1000 gm were included in the study”

8. Discussion: In this section: For the reader to follow, it is best to give the summary of key important findings in the first paragraph, and in the subsequent paragraphs to continue discussing your findings by comparing with other previously reported studies. It is also important to minimize repeating describing your findings in the discussion section. The discussion of your findings should be explicitly discussed since then all your results have been well elaborated in your results section. Eg. “The absence of partograph utilization observed in this study as one of the major determinant factors of OL was also reported in other studies in Ethiopia”. In addition, avoid reporting the odds ratio, percentages ( %), CI, P-Values in the discussion part. Just report in wording the interpretation of your findings and compare them with other previous studies. Prefer to speak, half of.., a third of.. rather than 50.3%, etc.

9. Some abbreviations have not been fully expressed at first use like “NICU”

10. Conclusion support the data presented in the study

11. For Table 3: (Determinants of obstructed labour in Hawassa Referral Hospital, Ethiopia); since it is long, no need for the subheading, just leave it for it to continue as it is.

12. Maintain continuity of the layout of all tables as instructed by the journal, Table 4 has a different layout compared with other tables.

13. It is advisable to follow the journal guidelines: For the PLOS one, the manuscript should be double spaced, with line numbers etc…

14. Revise your reference list to be consistent, eg. Number 50. The authors' list has small capitals.

6. PLOS authors have the option to publish the peer review history of their article (what does this mean?). If published, this will include your full peer review and any attached files.

Reviewer #1: No

Reviewer #2: No

---

## [Author Response · Author response to Decision Letter 0]

24 Nov 2021

Dear editors and reviewers 

We would like to extend our deepest appreciation for devoting your time to review our manuscript entitled " Determinants of obstructed labor and its adverse outcomes among women who gave birth in Hawassa University Referral Hospital: A case control study”. Globally, obstructed labour accounted for 12% of maternal death and most of the maternal morbidity and perinatal mortality. It is one of the most common preventable causes of maternal and perinatal morbidity and mortality in developing countries. 

Many of deaths and morbidities due to obstructed labour are entirely preventable and treatable. For this, the Ethiopian government had performed different activities. However, studies showed that adverse feto- maternal outcomes appear to be high and a common challenge in Ethiopia , the leading cause of maternal mortality in the with uterine rupture. Even though, the predictors associated with obstructed labour are scarce in Ethiopia. Determining the determinants of obstructed labor, maternal and perinatal outcomes of obstructed labour in our setting remains paramount to reduce the mortality and its morbidity. This study may be used for policy makers and a means of achieving the SDG target fetomaternal deaths from 2030. Therefore, this study was aimed to assess determinants of obstructed labor and its adverse outcomes of obstructed labor among women who gave birth in Hawassa teaching hospital, Southern Ethiopia. 

Dear reviewer, there has been a major revision of this manuscript (Abstract, introduction, methods, results discussion and conclusions). The language has been extensively examined to correct grammatical and spelling inconsistencies and the whole structure of the manuscript has been revised. We hope now the manuscript is clear and more acceptable than its previous version. We have tried to present the paper in proper manner according to your comment what to supposed to do so. For this, here we have given our responses to each of the concerns you raised, highlighted by red color. Again, we would like to remind our strongest gratitude for your effort for the improvement of this manuscript and all the points were addressed below. 

Regards 

Reviewer # 1 

General comment: authors should try and correct the spelling and grammatical errors.

Response: thank you for your scholarly comments and suggestions. 

Abstract: In the methods area mention should be made of the fact that the study was based on patient records/charts.

Response: accepted, the fact that the study was based on patient records/charts was putted.

Background: authors ought to refer to a similar study conducted by Mulugeta et al. 2020. Reasons as to why the study was done in Hawassa?

Response: Thanks for the scholarly comments. Actually, the paper published by Mulugeta D et al. was published in 2020, access in the year 2020, while our study was conducted in 2018. Again, there was difference in the study setting in which that was mulugeta’s was in Amhara region referral hospitals and ours was in Southern region. 

Discussion: Please authors include reference to a study by Mulugeta Dile et al. (2020) titled: Determinants of obstructed labour among women attending intrapartum care in Amhara Region NW Ethiopia: a hospital based unmatched case-control study.

Response: accepted and revision was made. 

In the limitations there is no mention about missing data which is common in use of secondary data.

Response: accepted and it is already included as “The retrospective nature of the study might prevent data collection for some variables like educational level, type of delays, socioeconomic status, nutritional status, and infrastructure of the health facilities as these variables were not registered in women’s obstetric cards.”

Conclusion: Strong recommendation on the use of family planning though relevant but does not come directly from the data.

Response: Accepted, but the recommendations is an area of implications based on the findings. Hence, low birth order is a prevention for obstructed labor, top reduce multiparity utilization of family planning is paramount and better to be recommended. 

Reviewer #2: 1. Generally, the article is very important to share with the wider community as it addresses determinants of obstructed labour which is one of the direct causes of maternal deaths. Knowing the determinants of obstructed labour will help to put measures that will improve maternal health and ultimately reduce maternal mortality.

Response: thank you for your interest on the area of interest. 

2. The abstract is more than 300 words based on journal guidelines, try to reduce redundant words.

Response: accepted and revision was made. 

3. There were a lot of grammatical errors throughout the documents. Needs to revise the whole document.

Response: Thank you very much and overall revision of the document was made. 

4. Background: The authors need to unpack and describe all the determinants of obstructed labour reported previously in other studies in the background section.

Response: accepted and remained determinants were incorporated. 

5. Ethic: In Ethics statement line 4, correct repeated word, “consent was”

Response: accepted and repeated word was removed. 

6. Sample size: The sample size calculation is not well computed; This is a case control study and decided to take a ratio of 1:2, how did you arrive with a total number of 486 women (162 women for cases and 324 women for controls)? This should be well elaborated.

Response: description of sample size calculation was narrated. 

7. Methods: In the Method section, under “Study design, setting and population” line 14...This sentence is not clear, “All women who gave birth after 28 weeks of gestation or weight of at least 1000 gm were included in the study”.

Response: we tried to explain to describe the deliveries after viability in Ethiopia context that is deliveries after 28 weeks or live births weights 1000 gm. 

8. Discussion: In this section: For the reader to follow, it is best to give the summary of key important findings in the first paragraph, and in the subsequent paragraphs to continue discussing your findings by comparing with other previously reported studies. It is also important to minimize repeating describing your findings in the discussion section. The discussion of your findings should be explicitly discussed since then all your results have been well elaborated in your results section. Eg. “The absence of partograph utilization observed in this study as one of the major determinant factors of OL was also reported in other studies in Ethiopia”. In addition, avoid reporting the odds ratio, percentages ( %), CI, P-Values in the discussion part. Just report in wording the interpretation of your findings and compare them with other previous studies. Prefer to speak, half of.., a third of.. Rather than 50.3%, etc.

Response: Thank you for the highly scholarly comments and revision was made. We have putting the interpretation of the findings and compare them with other previous studies and the odds ratio, percentages ( %), CI, P-Values in the discussion part is avoided. 

9. Some abbreviations have not been fully expressed at first use like “NICU”

Response: accepted and revision was made. 

10. For Table 3: (Determinants of obstructed labour in Hawassa Referral Hospital, Ethiopia); since it is long, no need for the subheading, just leave it for it to continue as it is.

Response: accepted and subheadings were removed from the Table 3. 

12. Maintain continuity of the layout of all tables as instructed by the journal, Table 4 has a different layout compared with other tables.

Response: accepted and revision was made. we have make the layout of Table 4 similar with other tables seems like. 

13. It is advisable to follow the journal guidelines: For the PLOS one, the manuscript should be double spaced, with line numbers etc…

Response: accepted and revision was made and line number was provided. 

14. Revise your reference list to be consistent, eg. Number 50. The authors' list has small capitals. 

Response: It is corrected and revised.

---

## [Editor Report · Decision Letter 1]

19 Jan 2022

PONE-D-21-03875R1

Determinants of obstructed labour and its adverse outcomes among women who gave birth in Hawassa University Referral Hospital: A case-control study

PLOS ONE

Dear Dr. Melaku Desta,

Thank you for submitting your manuscript to PLOS ONE. After careful consideration, we feel that it has merit but does not fully meet PLOS ONE’s publication criteria as it currently stands. Therefore, we invite you to submit a revised version of the manuscript that addresses the points raised during the review process.

We look forward to receiving your revised manuscript.

Kind regards,

Veneranda Masatu Bwana, MD, MSc, PhD

Academic Editor

PLOS ONE

Journal Requirements:

Additional Editor Comments (if provided):

1. Congrats to the authors for improving the manuscript. However, some of the comments from the reviews have not been satisfied.

2. The abstract is still more than 300 words. The abstract has a number of irrelevant words which could be omitted. Eg, in the Methods paragraph, words like… “ odds ratio/CI/ Statistical significance was declared when the p-value was less than 0.05” are not necessary to mention here. I will advise the authors to revisit the abstract again to be clear, short, and precise to comply with the journal requirements.

3. Some grammatical errors and tenses have not yet been satisfied, eg. “…sample size is …” instead of “…sample size was….”. Revise the whole document to clear all the minor spelling and tenses.
---

## [Author Response · Author response to Decision Letter 1]

4 Feb 2022

Dear editors of PLOS one 

We would like to extend our deepest appreciation for devoting your time to review our manuscript entitled " Determinants of obstructed labor and its adverse outcomes among women who gave birth in Hawassa University Referral Hospital: A case control study”. Globally, obstructed labour accounted for 12% of maternal death and most of the maternal morbidity and perinatal mortality. It is one of the most common preventable causes of maternal and perinatal morbidity and mortality in developing countries. 

Many of deaths and morbidities due to obstructed labour are entirely preventable and treatable. For this, the Ethiopian government had performed different activities. However, studies showed that adverse feto- maternal outcomes appear to be high and a common challenge in Ethiopia , the leading cause of maternal mortality in the with uterine rupture. Even though, the predictors associated with obstructed labour are scarce in Ethiopia. Determining the determinants of obstructed labor, maternal and perinatal outcomes of obstructed labour in our setting remains paramount to reduce the mortality and its morbidity. This study may be used for policy makers and a means of achieving the SDG target fetomaternal deaths from 2030. Therefore, this study was aimed to assess determinants of obstructed labor and its adverse outcomes of obstructed labor among women who gave birth in Hawassa teaching hospital, Southern Ethiopia. 

Dear reviewer, there has been a minor revision of this manuscript. The reference list has been extensively reviewed to ensure its completeness and to correct inconsistencies. We hope now the manuscript is clear and more acceptable than its previous version. We have tried to present the paper in proper manner according to your comment what to supposed to do so. For this, here we have given our responses to each of the concerns you raised, highlighted by red color. Again, we would like to remind our strongest gratitude for your effort for the improvement of this manuscript and all the points were addressed below. 

Regards

---

## [Decision Letter · Decision Letter 2]

12 May 2022

Determinants of obstructed labour and its adverse outcomes among women who gave birth in Hawassa University Referral Hospital: A case-control study

PONE-D-21-03875R2

Dear Dr. Desta,

We’re pleased to inform you that your manuscript has been judged scientifically suitable for publication and will be formally accepted for publication once it meets all outstanding technical requirements.

Kind regards,

Veneranda Masatu Bwana, MD, MSc, PhD

Guest Editor

PLOS ONE

Additional Editor Comments (optional):

Reviewers' comments:

Reviewer's Responses to Questions

**Comments to the Author**

1. If the authors have adequately addressed your comments raised in a previous round of review and you feel that this manuscript is now acceptable for publication, you may indicate that here to bypass the “Comments to the Author” section, enter your conflict of interest statement in the “Confidential to Editor” section, and submit your "Accept" recommendation.

Reviewer #3: All comments have been addressed

2. Is the manuscript technically sound, and do the data support the conclusions?

Reviewer #3: Partly

3. Has the statistical analysis been performed appropriately and rigorously? 

Reviewer #3: Yes

4. Have the authors made all data underlying the findings in their manuscript fully available?

Reviewer #3: Yes

5. Is the manuscript presented in an intelligible fashion and written in standard English?

Reviewer #3: No

6. Review Comments to the Author

Reviewer #3: Title: Determinants of obstructed labour and its adverse outcomes among women who gave birth in Hawassa University Referral Hospital: A case-control study

It is an interesting topic in developing countries.

Abstract:

Is ok. Except for the recommendation part that needs revision.

Background:

The last statement of the 1st paragraph … obstructed cases, must be corrected as Obstetric causes.

Second paragraph.. In Ethiopia, OL is associated with 3% of maternal death, 37.5% to 70% of perinatal deaths [9], and 36% of maternal death in Ethiopia. Why is this variation in the contribution of OL to maternal death? Which one is more likely true to cite?

The background is better rewritten as first the global perspective, then regional and national. In this case, it is not organized in this way. More recent literature on the title has to be included.

Method:

The study population is stated as… “Randomly selected women who gave birth in the last 3 years and fulfilled the inclusion criteria were the study population” how many OL was in the past three years to randomly collect the study population from?

How were the cases and controls identified? This must have been described in the data collection procedure part.

The criteria put under the definition of OL and clinical pelvimetry are not correct and confusing, and it has to follow only the standard definition that is put under the introduction.

Why were preterm included? It is less likely to have obstructed labor in preterm.

Result:

Why 156 out of 162? The reason has to be explained. Was the information incomplete or what?

Terms like more than half, two-third, etc, have to be avoided and the exact figure has to be described

The finding..More than one-fourth (28.2%) of cases had contracted pelvis.. is an unusual finding. It is overinflated.

The classification of gestational age in table 2 to <42 wks and > or =42 wks is not appropriate. Further classification is needed. You may consider <37 wks, 37 to 42 wks, and >42 wks. Similarly, you have to choose one classification for birth weight. The latter is appropriate (Table3). Do all the women know their gestational age or LNMP? You didn’t use another surrogate to estimate gestational age.

What would be the explanation for the finding…. Primiparous women were 81% times [AOR= 0.19, 95% CI: 0.07, 0.52] and multigravida women were 83% times [AOR = 0.17, 95 % CI: 0.07, 0.41] less likely to have OL than grand multiparous women. Is primigravida not at risk of having OL than multigravida?

Another query..Similarly, women who had contracted pelvis were about 4 times more likely to have the chance of obstructed labour than those who had adequate pelvis [AOR = 3.98, 95% CI: 1.68, 9.42].. Do you mean that there are no women with contracted pelvis who didn’t develop OL? Women with contracted pelvis will have 100% OL if it is not intervened timely. That is why I said the diagnosis of the contracted pelvis is overinflated. Could there be a diagnosis difficulty, the skill to diagnose??

What about those far than 50 KMs? …. The odds of OL were 3.89 times more likely among women who reside within 10-50 kilometers estimated distance from the facility than those who reside below 10 kilometers distance [AOR= 3.89, 95% CI: 1.14, 13.3]. you may need to reclassify as >/=10KMs and do the analysis again.

How was it possible to get the distance in KMs from the retrospective review?

Discussion:

This discussion in the first paragraph needs correction….A study conducted in LMICs [31] also showed that gravidity ≥ 2 was protective of OL. This variation might be due to socio-demographic differences and the lower risk of malpresentation and malposition among primigravida women… We rather expect more malpresentation and malposition in primigravida.

The discussion part needs revision and the content must be seen carefully.

Conclusion:

Must be shortened and specific.

7. PLOS authors have the option to publish the peer review history of their article (what does this mean?). If published, this will include your full peer review and any attached files.

Reviewer #3: No

---

## [Editor Report · Acceptance letter]

14 Jun 2022

PONE-D-21-03875R2 

Determinants of obstructed labour and its adverse outcomes among women who gave birth in Hawassa University Referral Hospital: A case-control study 

Dear Dr. Desta:

I'm pleased to inform you that your manuscript has been deemed suitable for publication in PLOS ONE. Congratulations! Your manuscript is now with our production department. 

Kind regards, 

on behalf of

Dr. Veneranda Masatu Bwana 

Guest Editor

PLOS ONE